# Serum susceptibility of *Escherichia coli* and its association with patient clinical outcomes

**Orianna Poteete**[1], **Phillip Cox**[1], **Felicia Ruffin**[1], **Granger Sutton**[2], **Lauren Brinkac**[2], **Thomas H. Clarke**[2], **Derrick E. Fouts**[2], **Vance G. Fowler, Jr.**[1,3], **Joshua T. Thaden**[1]*

**1** Department of Medicine, Division of Infectious Diseases, Duke University School of Medicine, Durham, NC, United States of America, **2** J. Craig Venter Institute, Rockville, MD, United States of America, **3** Duke Clinical Research Institute, Durham, NC, United States of America

* Joshua.Thaden@duke.edu

## Abstract

The innate immune system eliminates bloodstream pathogens such as *Escherichia coli* in part through complement protein deposition and subsequent bacterial death (i.e., "serum killing"). Some *E. coli* strains have developed mechanisms to resist serum killing, though the extent of variation in serum killing among bloodstream infection (BSI) isolates and the clinical impact of this variation is not well understood. To address this issue, we developed a novel assay that uses flow cytometry to perform high throughput serum bactericidal assays (SBAs) with *E. coli* BSI isolates (n = 183) to define the proportion of surviving bacteria after exposure to serum. We further determined whether *E. coli* resistance to serum killing is associated with clinical outcomes (e.g., in-hospital attributable mortality, in-hospital total mortality, septic shock) and bacterial genotype in the corresponding patients with *E. coli* BSI. Our novel flow cytometry-based SBA performed similarly to a traditional SBA, though with significantly decreased hands-on bench work. Among *E. coli* BSI isolates, the mean proportion that survived exposure to 25% serum was 0.68 (Standard deviation 0.02, range 0.57–0.93). We did not identify associations between *E. coli* resistance to serum killing and clinical outcomes in our adjusted models. Together, this study describes a novel flow cytometry-based approach to the bacterial SBA that allowed for high-throughput testing of *E. coli* BSI isolates and identified high variability in resistance to serum killing among a large set of BSI isolates.

## Introduction

*Escherichia coli* is one of the most widely studied bacterial species [1,2]. *E. coli* has significant variability in its genetic repertoire, with genetic elements that promote colonization, disruption of epithelial barriers, and tissue invasion [3]. Invasion of *E. coli* into the bloodstream leads to bacteremia, sepsis, and a high risk of death [4–6]. Our first line of defense against *E. coli* in the bloodstream is the complement system, a component of innate immunity [7]. The complement system is a regulated network of proteins in the blood that is composed of three pathways: classical, lectin, and alternative. Each pathway is activated through different means, such

**Data Availability Statement:** There are legal and ethical restrictions on sharing patient-level data. Sharing of clinical data requires review and approval of a Data Use Agreement by the Duke

Institutional Review Board. Request for access to this data can be made by contacting Vance Fowler (vance.fowler@duke.edu) and the Duke University Health System IRB office (919-668-5111).

**Funding:** J.T.T.; National Institute of Allergy and Infectious Diseases, National Institutes of Health; K08 AI171183; https://www.niaid.nih.gov/ V.G.F.; National Institute of Allergy and Infectious Diseases, National Institutes of Health; R01 AI165671; https://www.niaid.nih.gov/ D.E.F.; Department of Health and Human Services; U19AI110819; https://www.hhs.gov/ D.E.F.; Centers for Disease control and Prevention (CDC) Epicenter Program; U54CK000603; https://www.cdc.gov/hai/epicenters/index.html. In no case did the funders play any role in the study design, data collection and analysis, decision to publish, or preparation of the manuscript.

**Competing interests:** The authors have declared that no competing interests exist.

as binding of antibody, mannose-binding lectin, or C3b to bacteria, respectively [7–11]. Activation of any pathway can result in formation of the membrane attack complex (MAC), a pore in the bacterial plasma membrane that triggers pathogen death through osmolysis [7,9,12]. Given that bacteria are killed by complement proteins within the blood (or serum), this process is referred to as *serum killing* [13].

While the human complement system is designed to eliminate bacteria from the bloodstream, some *E. coli* strains have evolved mechanisms to resist killing (i.e., serum resistance) [13,14]. For example, the presence of capsular polysaccharides may prevent MAC binding and secretion of proteases may degrade complement proteins [13–16]. However, the degree to which clinical strains of *E. coli* are resistant to serum killing, and the impact of this resistance on patient outcomes, is unknown. One of the challenges in addressing this issue is the extensive time and labor needed to perform traditional serum bactericidal assays (SBAs) with large collections of bacterial isolates [17]. To address this challenge, we describe in this report the development and validation of a flow cytometry-based high throughput SBA. We used this novel approach to define the variability in resistance to serum killing within a large set of *E. coli* BSI isolates, and then determined the extent to which this resistance is associated with patient clinical outcomes (e.g., septic shock, total in-hospital mortality, attributable in-hospital mortality) and bacterial genotype.

## Methods

### Study population and definitions

The patient clinical data and bacterial isolates were obtained from the Duke Blood Stream Infection Biorepository (BSIB). The BSIB contains prospectively collected clinical data and bacterial BSI isolate from >4000 unique adult inpatients at Duke University Health System with monomicrobial Gram-negative bacterial BSI since 2002. The patients in this study were enrolled between January 1, 2002 and December 31, 2015. The clinical data was accessed on January 5, 2023. Written informed consent was obtained from all study participants or their legal representatives, and the study was approved by the Duke University Institutional Review Board. Hospital-acquired infection was defined as infection beginning ≥48 hours after hospital admission [18]. Community-acquired bloodstream infection was defined as an infection beginning <48 hours after hospital admission. Community-acquired bloodstream infection was further subdivided into the following: 1) community-acquired, healthcare-associated bloodstream infection, and 2) community-acquired, non-healthcare-associated bloodstream infection. The community-acquired, healthcare-associated bloodstream infection definition was modified from Friedman *et al.* [18], and defined as a bloodstream infection beginning prior to 48 hours after hospital admission in patients that meet one or more of the following criteria: hospitalized in the past 90 days, resident of a nursing home or long-term care facility, actively receiving home intravenous therapy, received wound care or specialized nursing care in previous 30 days, received hemodialysis in past 30 days, immunosuppressed (e.g., presence of metastatic cancer, history of a solid organ or hematological transplant, chemotherapy in last 30 days, currently on immunosuppressive medication for any reason), or surgery in last 180 days. Community-acquired, non-healthcare-associated bloodstream infection is any community-acquired bloodstream infection not meeting the criteria for healthcare-associated bloodstream infection. The *source* of infection refers to the primary focus of the bloodstream infection (e.g., urine/pyelonephritis, line, *etc.*). Acute Physiology and Chronic Health Evaluation II (APACHE-II) score [19] was calculated at the time of initial positive blood cultures. In-hospital mortality was defined as death prior to hospital discharge. Recurrent bloodstream infection was defined as present if there was a clinical and microbiological resolution of the

initial episode of infection after treatment, but culture-confirmed *E. coli* (same antibiogram) was documented within the index hospital admission. Complications of *E. coli* bloodstream infection including septic shock, acute kidney injury (AKI), acute lung injury / acute respiratory distress syndrome (ALI/ARDS), and disseminated intravascular coagulation (DIC) were defined per standard guidelines [20–23]. Appropriate antibiotic therapy is defined as the receipt of an antibiotic to which the bacteria is susceptible. Appropriate antibiotic therapy was determined daily from the date of the index positive blood culture (day 0) to hospital discharge or death. Antimicrobial susceptibility testing was performed by the Duke Clinical Microbiology Lab using standard techniques. The multidrug resistant (MDR) phenotype was defined as in Magiorakos et al. [24].

## Traditional serum bactericidal assay (SBA)

As in the standard approach for performing an SBA, *E. coli* isolates were streaked onto LB plates and incubated at 37˚C overnight (ON). One milliliter LB cultures were inoculated from single colonies and incubated at 37˚C ON while shaking at 225 rpm. The cultures were spun down at 13,000xg for 3 minutes and the cells washed twice with PBS. The cultures were diluted with PBS to $OD_{600}$ of 0.10. The serum (0%, 25%, 50%) was prepared using pooled human serum (Sigma-Aldrich, catalog number H4522) and PBS. Then, 180 μL of serum (0%, 25%, or 50%) and 20 μL of the bacterial isolate were mixed by pipetting in a 96-well plate. Plates were incubated at 37˚C for 30 minutes and centrifuged at 3,000 rpm for 20 minutes. Cells were washed twice with PBS and serially diluted 1000-fold with PBS. Fifty microliters of the diluted samples were plated on LB and incubated ON at 37˚C. Colony forming units (CFU) were counted and recorded. The proportion of living colonies for both 25% serum and 50% serum samples were calculated using the formulas below:

$$proportion\ live\ colonies\ in\ 25\%\ serum = \frac{of\ CFU\ in\ 25\%\ serum}{of\ CFU\ in\ 0\%\ serum}$$

$$proportion\ live\ colonies\ in\ 50\%\ serum = \frac{of\ CFU\ in\ 50\%\ serum}{of\ CFU\ in\ 0\%\ serum}$$

## Novel flow cytometry-based SBA

The initial steps of our novel flow cytometry-based SBA, including culture growth, washes, dilution, incubation with serum, and subsequent washes with PBS were identical to the traditional SBA protocol described above. In this new approach, the bacteria were then resuspended in PBS and transferred to FACS tubes. The live/dead stain 7-Aminoactinomycin D (7-AAD; Invitrogen, catalog number A1310) was added at a final concentration of 40 μg/mL. The samples were analyzed with a BD Biosciences FACS Canto machine. In total, 10,000 events were recorded using an excitation wavelength of 488 nm and a detection wavelength of 647 nm. The recorded values were used to produce the proportion of living cells after exposure to 25% or 50% serum using the formulas described above. For assay validation, four biological replicates across two experiments were performed. Given the low variance noted in the validation assay, serum susceptibility assays were performed in duplicate on the remaining BSI isolates. An overall schematic of the traditional and flow cytometry-based SBA approaches is shown in **Fig 1**.

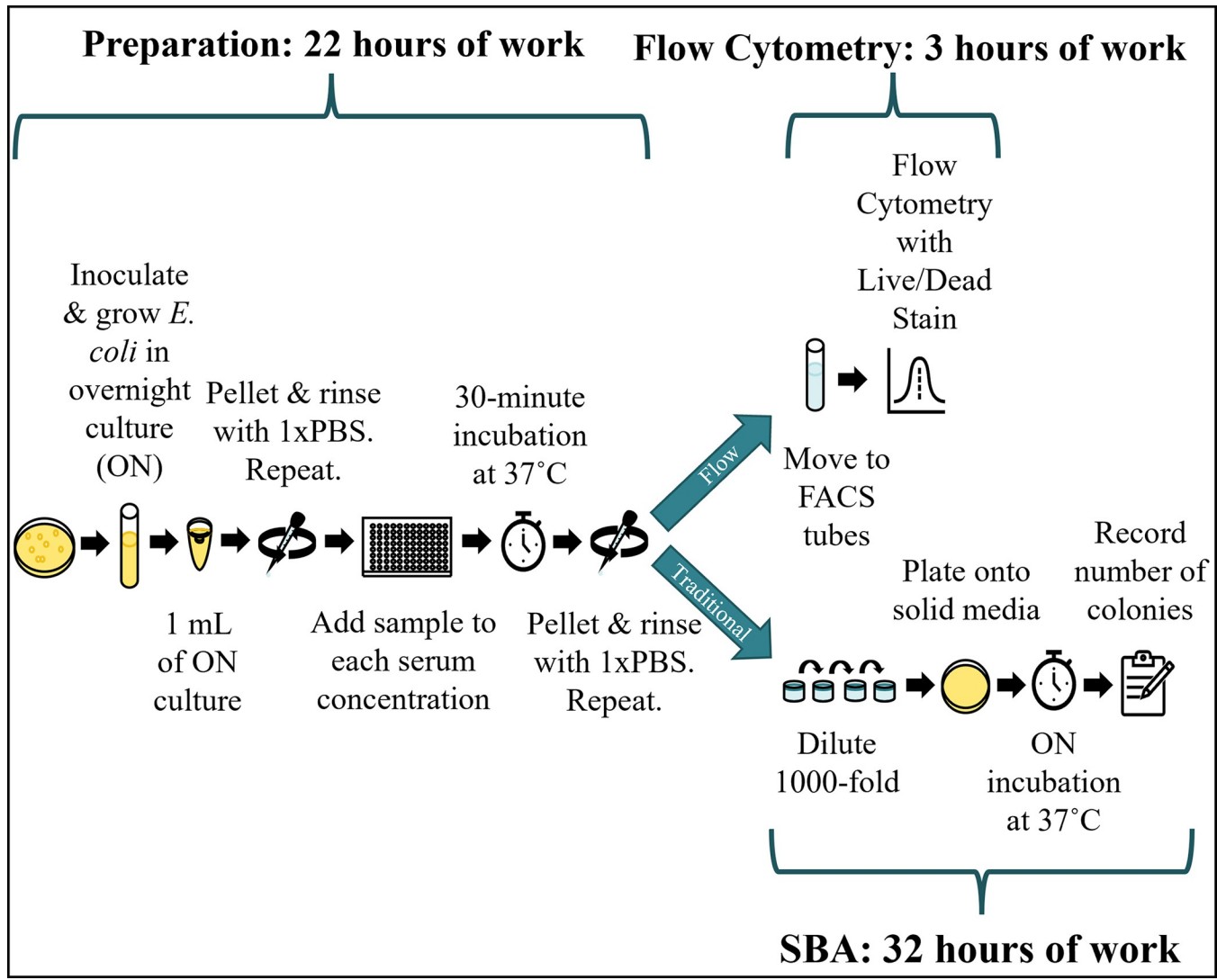

**Fig 1. Workflow and duration differences between traditional serum bactericidal assay (SBA) and flow cytometry-based SBA methods.** Both the traditional and flow-based methods begin the same way. After *E. coli* strains are incubated with serum and washed with PBS, the two approaches diverge (blue arrows). The blue bars indicate the approximate duration for a single person to perform an experiment with a full 96-well plate of samples. This corresponds to 16 *E. coli* strains exposed to 0%, 25%, and 50% serum, performed in duplicate. Abbreviations: FACS, fluorescent activated cell sorting; ON, overnight; PBS, phosphate-buffered saline.

### Statistical analyses

In the description of patient and bacterial characteristics, continuous variables were reported as means with standard deviations. Dichotomous variables were reported as counts and percentages. T-tests were used to examine associations between *E. coli* resistance to serum killing and clinical outcomes or bacterial sequence type. Two-way analysis of variance (ANOVA) tests were used to validate our novel flow cytometry-based method relative to the traditional SBA. This test accounted for both the bacterial isolate (high, medium, and low serum susceptibility isolates were tested) and the method for determining serum susceptibility (flow cytometry-based versus traditional SBA). Chi-square tests were used to broadly identify differences in serum susceptibility among *E. coli* phylogroups and sequence types, and t-tests were used to identify differences in susceptibility between particular genetic groups (e.g., ST131) and all

others. For the serum susceptibility histogram, a Gaussian line of best fit was generated using a nonlinear regression approach (GraphPad Prism, Boston, MA). In the adjusted analyses, logistic regression models were generated to determine associations. Adjusted models were generated for clinical outcomes including total in-hospital mortality, attributable in-hospital mortality (i.e., death due to infection as opposed to other causes), and septic shock. Model covariates included age, gender, race, route of infection (e.g., hospital-acquired infection, etc.), source of infection, hematopoietic or solid organ transplant, diabetes mellitus, recent corticosteroid use (within 30 days prior to BSI), HIV, recent surgery (within 30 days prior to BSI), days to effective antibiotics, chronic health APACHE-II score, and *E. coli* resistance to serum killing. *E. coli* resistance to serum killing for each BSI isolated was represented as the proportion of live bacteria after exposure to 25% serum relative to the 0% serum control. Therefore, this variable could range from 0 (fully killed by 25% serum) to 1 (fully resistant to serum killing). The chronic health portion of the APACHE-II describes whether a patient has severe organ system insufficiency on hospital admission. These covariates were selected to broadly encompass the clinical factors known or thought to influence BSI outcome. Model covariates with near significant p-values ($p \leq 0.15$) in a univariable analysis were included in the final multivariable logistic regression models. P-values less than 0.05 were considered significant.

The traditional SBA (i.e., non-flow cytometry-based SBA) produced considerable variation between replicates. Therefore, data analysis was performed both with and without formal outlier removal. The data presented here involved outlier removal from the traditional SBA samples, though the overall results and findings did not differ whether outliers were removed or not. The interquartile range (IQR) technique was used for outlier removal [25]. No outlier removal was necessary for the flow cytometry-based SBA given homogeneity of results across replicates.

## Whole genome sequencing and assembly

The *E. coli* genomic sequencing data used in this study was similarly used in a prior study [26]. DNA isolation, library construction, and assembly was described in this prior study. In brief, multilocus sequence typing (MLST) was performed using LOCUST [27]. MLST was performed by using the whole genome sequence data for each *E. coli* BSI isolate to determine the *adk*, *fumC*, *gyrB*, *icd*, *mdh*, *purA*, and *recA* alleles present in each isolate. The pattern of alleles in each isolate was then matched to the particular sequence type (ST) using a standard database [28]. Pan-genome analysis was performed through the JCVI pan-genome pipeline using the Pan-genome Ortholog Clustering Tool (PanOCT) [29,30]. *E. coli* phylogroups were identified through either known associations with ST (e.g., ST131 and phylogroup B2) or through the *In Silico* Clermont Phylotyper [31]. All genomes used in this study are available at NCBI under BioProject number PRJNA290784.

## Results

### Creation and validation of high throughput serum susceptibility assay

While the usual method for determining serum susceptibility is through SBAs, this method is unwieldy when used for a large number of isolates due to the need for diluting, plating, and counting colonies for each experimental sample [32,33]. Therefore, we adapted a protocol from Khan et. al. to develop a high-throughput flow cytometry-based technique to measure serum susceptibility in *E. coli* [34–36]. The overall schematic of time needed to complete the traditional and flow-cytometry based SBAs is shown in **Fig 1**, and further detailed in **S1 Fig**. The novel flow cytometry-based SBA decreased overall experiment duration (25 versus 54 hours) and hands-on benchwork time (8 versus 20 hours) per 16 *E. coli* BSI isolates (i.e., one

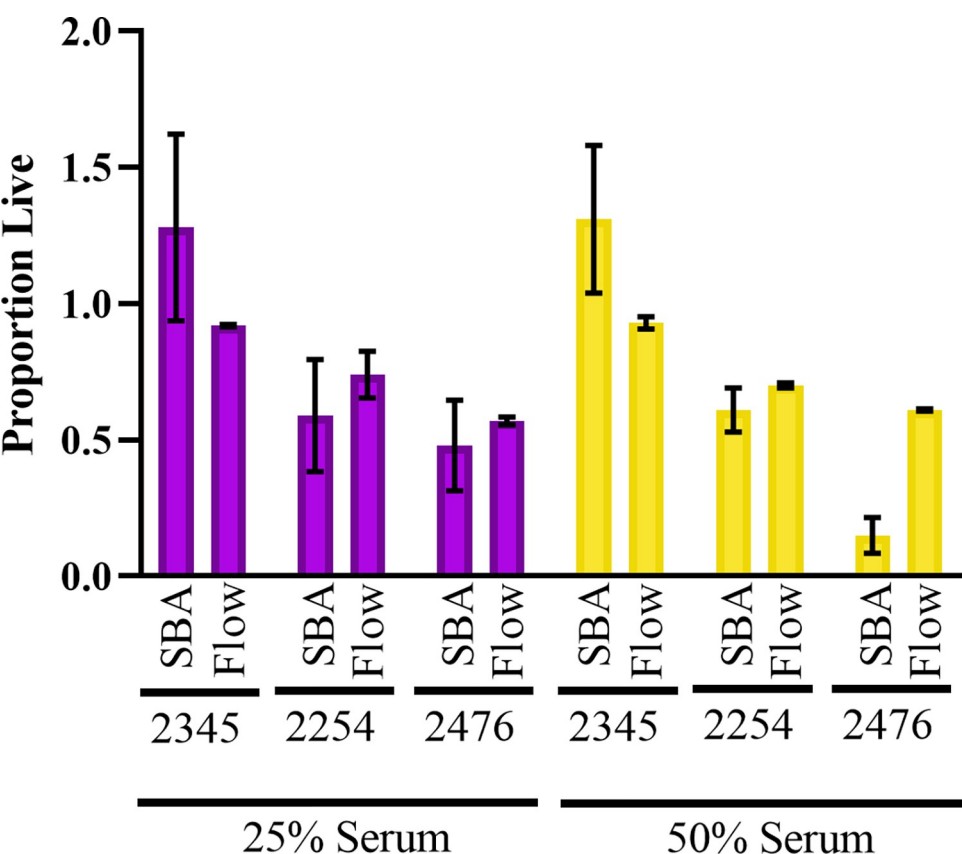

**Fig 2. Validation of flow-cytometry-based serum bactericidal assay (SBA).** The serum susceptibility of *E. coli* strains 2345, 2254, and 2476 was determined through both a traditional SBA (SBA) and our novel flow cytometry-based (Flow) approach. The number of live and dead bacteria was determined in the presence of 0%, 25%, and 50% serum. In each case, the proportion of surviving bacteria (i.e., the number of live bacteria in serum relative to the serum-free control) is plotted. Relative differences in serum susceptibility between the *E. coli* strains (e.g., low, medium, and high susceptibility) were the same with the two experimental approaches (25% serum: p = 0.76; 50% serum: p = 0.79). The traditional SBA experiment included five biological replicates across two experiments. The flow cytometry-based assay included four biological replicates across two experiments.

96-well plate in the flow cytometry-based SBA). The novel flow cytometry-based SBA successfully identified trends in serum susceptibility relative to the "gold standard" traditional SBA (**Fig 2**). Specifically, *E. coli* strains 2345, 2254, and 2476 exhibited high, medium, and low resistance to serum killing by the traditional SBA, respectively, which was similarly ordered by the novel flow cytometry-based SBA. There were no statistically significant differences in serum susceptibility between the two methodologies when isolates were exposed to either 25% serum (p = 0.76) or 50% serum (p = 0.79).

### Resistance to serum killing among *E. coli* BSI isolates

In total, 183 patients with *E. coli* BSI were included in this study. To investigate the serum susceptibility of these isolates, we used our novel flow-based SBA approach to determine the proportion of surviving bacteria after exposure to serum (**Fig 3**). The mean proportion that survived after exposure to 25% serum, relative to 0% serum, was 0.68 (standard deviation 0.02). The mean proportion that survived after exposure to 50% serum, relative to 0% serum, was 0.67 (standard deviation 0.02). While most of the *E. coli* BSI isolates demonstrated serum survival within a relatively narrow range (0.60–0.75 proportion survival), there were outliers

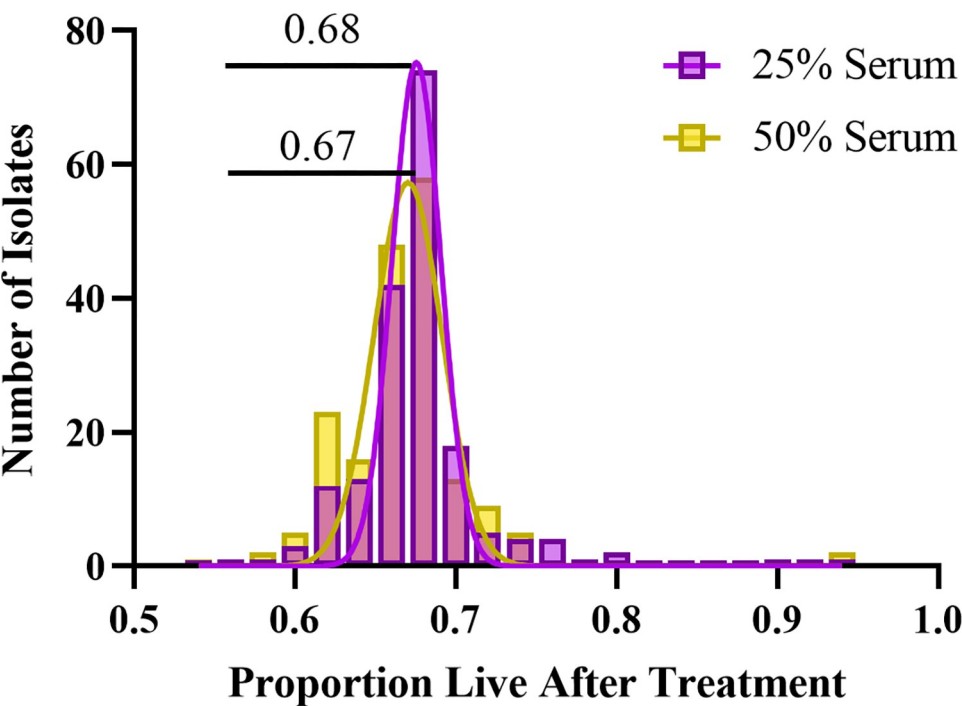

**Fig 3. Distribution of susceptibility to serum killing among *E. coli* bloodstream infection (BSI) isolates.** *E. coli* BSI isolates (n = 183) were treated with either 25% serum (purple) or 50% serum (yellow), and the number of surviving bacteria relative to a 0% serum control was determined. The distributions of the proportion of surviving bacteria are shown.

that were both more resistant to serum (n = 10 isolates had >0.75 survival after exposure to 25% serum) and less resistant to serum (n = 2 isolates had <0.60 survival after exposure to 25% serum).

### Association between *E. coli* resistance to serum killing and patient clinical outcomes

The clinical characteristics and outcomes of the 183 patients with *E. coli* bloodstream infection are shown in **Table 1**. The patients had a mean age of 63 years (SD 15 years), and 87 (48%) were female. The most common source of BSI was the urinary tract (84/183 [46%]). There was significant antibiotic resistance as 123 (67%) were multidrug resistant. Overall outcomes were poor with 20% total in-hospital mortality (37/183), 13% attributable in-hospital mortality (24/183), and 25% of patients experiencing septic shock (45/183). We did not identify any significant associations between serum susceptibility and the patient and bacterial variables in Table 1.

We sought to determine if *E. coli* resistance to serum killing was associated with the clinical outcomes of total in-hospital mortality, attributable in-hospital mortality, or septic shock. We hypothesized that increased survival in 25% serum would be associated with increased rates of these patient complications. In unadjusted analyses (i.e., t-tests), however, the only association we identified was decreased attributable mortality with increased resistance to serum killing (Survival 0.68 in those that survived or died of other causes versus 0.67 in those that died due to BSI; p = 0.04). No associations between *E. coli* resistance to serum killing (25% serum) and total mortality (Serum survival 0.68 in those that survived to discharge versus 0.67 in those

**Table 1. Characteristics and outcomes of patients with *E. coli* bloodstream infections (BSI).** Abbreviations: HIV, human immunodeficiency virus; SD, standard deviation.

| Patient characteristics | n (%) N = 183 |
|---|---|
| **Age in years (mean [SD])** | 63 (15) |
| **Female** | 87 (48) |
| **Race** | |
| White | 119 (65) |
| Black | 48 (26) |
| Other | 16 (9) |
| **Medical comorbidities** | |
| Transplant (solid organ or hematopoietic stem cell) | 22 (12) |
| Diabetes mellitus | 65 (36) |
| Surgery past 30 days | 47 (26) |
| Hemodialysis | 18 (10) |
| Corticosteroid use in past 30 days | 40 (27) |
| HIV | 2 (1) |
| **Acute APACHE II score (mean [SD])** | 8 (6) |
| **Chronic APACHE II score (mean [SD])** | 4 (2) |
| **Source of BSI** | |
| Urine/pyelonephritis | 84 (46) |
| Biliary tract | 15 (8) |
| Line | 5 (3) |
| Abscess | 6 (3) |
| Pneumonia | 8 (4) |
| Skin / soft tissue | 10 (5) |
| Other | 23 (13) |
| None identified | 32 (17) |
| **Route of infection** | |
| Hospital-acquired | 35 (19) |
| Community acquired/Healthcare-associated | 116 (63) |
| Community-acquired/Non-healthcare-associated | 32 (17) |
| **Days to appropriate antibiotics** | |
| 0 days | 111 (61) |
| 1 day | 38 (21) |
| 2 days | 17 (9) |
| ≥3 days | 15 (8) |
| Unknown | 2 (1) |
| **Patient outcomes** | n (%) N = 183 |
| **Total in-hospital mortality** | 37 (20) |
| **Attributable in-hospital mortality** | 24 (13) |
| **Complications of BSI** | |
| Septic shock | 45 (25) |
| Acute kidney injury | 60 (33) |
| Acute lung injury / acute respiratory distress | 5 (3) |
| **Bacterial characteristics** | n (%) N = 183 |
| **Fluoroquinolone resistant** | 89 (49) |
| **Multidrug resistant (MDR)** | 123 (67) |
| **Extensively drug resistant (XDR)** | 0 (0) |

that died prior to discharge; p = 0.27) or septic shock (Serum survival 0.68 in those that did not develop septic shock versus 0.67 in those that did; p = 0.16) were identified.

Adjusted models of total in-hospital mortality and attributable in-hospital mortality are shown in **S1** and **S2** **Tables**, respectively. Clinical factors associated with increased total mortality in this cohort included increasing age (Odds ratio 1.04; 95% confidence interval 1.01–1.08; p = 0.02), community-acquired/healthcare-associated infection (relative to hospital-acquired infection) (Odds ratio 0.36; 95% confidence interval 0.13–0.99; p = 0.05), hemodialysis dependence (Odds ratio 5.43; 95% confidence interval 1.61–18.92; p<0.01), and an unknown source of BSI (Odds ratio 4.65;83 95% confidence interval 1.53–14.; p<0.01). Given that these multivariable logistic regression models involved only covariates that were significant or near significant in univariable analyses (i.e., p<0.15), *E. coli* resistance to serum killing was only included in the attributable mortality model; however, it was not significant (OR < 0.01, 95% CI <0.01–86.19, p = 0.26). An adjusted model of septic shock was not generated as no covariates demonstrated significance in univariable analyses.

## Associations between *E. coli* resistance to serum killing and bacterial genetics

The phylogenetic relationships among the *E. coli* BSI isolates included in this study, as well as the associated resistance to serum killing and clinical outcomes metadata for each BSI isolate, is shown in **S2 Fig**. *E. coli* phylogroups included A (n = 3 [2%]), B1 (n = 7 [4%]), B2 (n = 129 [70%]), C (n = 1 [1%]), D (n = 37 [20%]), and F (n = 6 [3%]). Overall, there was no difference in serum killing among phylogroups (25% serum: p = 0.27; 50% serum: p = 0.59). Similarly, pairwise comparisons (e.g., phylogroup B2 vs. all others) did not identify phylogroups associated with higher or lower serum killing.

In total, 36 *E. coli* STs were identified among the BSI isolates in this study. The *E. coli* STs with ≥5 representatives in our study population, along with the mean and range of surviving bacteria after serum exposure, are shown in **Table 2**. Interestingly, the only ST that statistically differed in its resistance to serum killing was ST131, which had slightly lower resistance to serum killing relative to non-ST131 strains (ST131: mean proportion surviving in 25% serum 0.67 [SD 0.026]; non-ST131: mean proportion surviving in 25% serum 0.68 [SD 0.061]; p = 0.05).

## Discussion

The variability in serum susceptibility among clinical bacterial strains and its impact on the outcomes of patients is an underexplored area of research. This is an important gap as

**Table 2. Resistance to serum killing among *E. coli* bloodstream infection isolates, stratified by multilocus sequence type (ST).** A serum bactericidal assay was performed on each *E. coli* isolate, and survival was determined by number of colonies in the serum-treated sample (either 25% or 50% serum) divided by the number of colonies in the negative serum control. The mean survival and range of survival values for each *E. coli* ST with ≥5 isolates in our study population are shown here. P-values were calculated with t-tests (e.g., proportion of surviving bacteria in ST12 relative to non-ST12 isolates). P-values ≤0.05 are in bold.

| ST | # isolates | 25% serum | | | 50% serum | | |
|---|---|---|---|---|---|---|---|
| | | Mean proportion surviving | Range of proportion surviving | P-value | Mean proportion surviving | Range of proportion surviving | P-value |
| **12** | 7 | 0.72 | 0.65–0.90 | 0.21 | 0.71 | 0.66–0.89 | 0.20 |
| **69** | 12 | 0.70 | 0.65–0.93 | 0.41 | 0.69 | 0.61–0.94 | 0.41 |
| **73** | 12 | 0.68 | 0.61–0.77 | 0.97 | 0.67 | 0.58–0.72 | 0.88 |
| **95** | 18 | 0.68 | 0.62–0.92 | 0.84 | 0.68 | 0.59–0.93 | 0.60 |
| **127** | 5 | 0.68 | 0.66–0.70 | 1.00 | 0.67 | 0.64–0.71 | 1.00 |
| 131 | 67 | 0.67 | 0.61–0.74 | **0.05** | 0.66 | 0.60–0.73 | **0.05** |
| **393** | 10 | 0.66 | 0.57–0.76 | 0.17 | 0.66 | 0.60–0.74 | 0.41 |
| **405** | 6 | 0.67 | 0.64–0.70 | 0.29 | 0.66 | 0.62–0.68 | 0.50 |

complement-mediated serum killing is the host's first line of defense against pathogens that invade the bloodstream, and deficiencies in the complement system are associated with recurrent and severe infections [7,9,10,12,37,38]. While the mechanistic basis of the complement system has been well-studied, less is known about bacterial variability in susceptibility to complement-mediated serum killing and how susceptibility impacts the clinical outcomes of patients with bacterial infections. The lengthy and tedious nature of the current methodologies (i.e., SBAs) for performing high-throughput analyses of bacterial serum susceptibility complicates our ability to address these questions [32,33,36]. We hope that the development of a high throughput method for determining bacterial serum susceptibility, as described here, could pave the way for additional large studies into this variability. This study had three major findings, which are discussed in detail below.

First, our novel flow cytometry-based approach for measuring *E. coli* serum susceptibility produced results similar to the gold standard study (traditional SBA) but with decreased overall duration, hands-on bench work, and variation between biological replicates. This work extends upon prior methodological work. Fluorescent stains, coupled with flow cytometry, have been used to assess the dynamics of bacterial death secondary to irradiation, isopropanol, or antibiotics [39–43]. Luminescence-based strategies to measure bacterial killing (including complement-mediated killing) have been published, though this approach is impractical for studying large numbers of bacterial strains given the requirement for introducing a luciferase gene into the bacterial strains of interest [44–46]. Prior work has also increased the throughput of SBAs through automated colony counting [47,48], or by detecting ATP release from dead/ dying bacteria [49–53]. Here we demonstrate a technique that avoids the need for any plating and colony counting yet retains the ability to count individual viable bacteria. One prior study used a fluorescent live/dead stain (propidium iodide) to assay bacterial viability following exposure to serum [54], though examined only a single bacterial strain and did not scale the approach to investigate multiple isolates.

Second, we utilized the novel flow cytometry-based SBA method to demonstrate high variation in serum resistance (~50–100% live bacteria after treatment) among *E. coli* bloodstream infection isolates. To our knowledge, no prior studies have systematically described the serum susceptibility of a large set of clinical *E. coli* BSI isolates. One study assayed serum killing among a set of 20 *E. coli* BSI isolates [55], while another similarly assayed serum killing among 20 *E. coli* BSI isolates as a control group for *E. coli* isolated from infected orthopedic hardware [56]. Other studies examined serum susceptibility of *E. coli* clinical isolates from the urinary and gastrointestinal tracts [56–58]. These prior studies have also demonstrated significant variation in resistance to serum killing among clinical *E. coli* isolates. These studies identified *E. coli* strains that had higher serum sensitivity than those in this study (e.g., as low as 1% survival relative to serum-free controls), though these prior studies differed in both the serum used (75% normal human serum from single donor) and incubation time (1–3 hours).

Third, there were no significant associations between resistance to serum killing and the clinical outcomes of interest (total in-hospital mortality, attributable in-hospital mortality, septic shock) in our adjusted models. Patient outcomes depend on a complex interplay of patient, treatment, and pathogen variables, and the impact of *E. coli* resistance to serum killing on patient outcomes may not be particularly significant. However, it is challenging to fully assess the impact of resistance to serum killing in the absence of patient-specific serum samples which may vary in their complement levels and activity. In a rabbit model of infective endocarditis, resistance to killing in rabbit serum was shown to be an important factor in the ability of *E. coli* to generate a persistent infection in this organism [59].

This study has several limitations. First, all the isolates came from a single geographic area, so some STs that are less common here will not be well represented. Second, we used

commercially available pooled human serum and not serum from the original patient. There could be differences in serum susceptibility to the corresponding patient's serum versus that observed here using pooled human serum. These differences could have contributed to the lack of association between serum resistance and patient clinical outcomes. Finally, this work does not capture additional important actions of the complement system such as opsonin-guided phagocytic activity or modulation of cytokine release.

In conclusion, complement-mediated killing of bacterial pathogens is a critical component of the human innate immune defense against bloodborne bacteria, though variation in bacterial resistance to serum killing, and its impact on clinical outcomes, is an underexplored area of study. This gap in the scientific literature is in part related to the lengthy and tedious nature of the assays traditionally used to examine resistance to serum killing among bacterial strains. To address this issue, we here described the development of a novel high-throughput, flow cytometry-based SBA that can be used to probe variations in complement-mediated killing of bacterial strains. We used this novel assay to probe resistance to complement-mediated serum killing in a large set of *E. coli* BSI isolates. We identified variation in resistance to serum killing among these BSI isolates, though did not identify associations between this resistance and clinical outcomes of mortality and septic shock. Future work should focus on examinations of the mechanisms of resistance to complement-mediated serum killing and its impact on clinical outcomes in other settings. While we did not identify associations between resistance to serum killing and clinical outcomes in our cohort, such associations have been identified in an animal model of bloodstream infection [59].

## Supporting information

**S1 Fig. Comparison of time requirements (hours) for traditional serum bactericidal assay (SBA) and novel flow cytometry-based SBA.** The estimates are time required for a single person to perform the experiment with 16 *E. coli* strains (0%, 25%, and 50% serum) in duplicate. For the flow cytometry-based SBA, this corresponds to one full 96-well plate.
(TIF)

**S2 Fig. Phylogenetic analysis of *E. coli* bloodstream infection isolates with associated resistance serum susceptibility and clinical outcomes data.** The innermost two colored range indicate the most common multilocus sequence types and phylogroup, respectively. The outer colored strips indicate the live bacteria proportions associated with exposure to 25% serum (yellow) and 50% serum (purple) as well as the corresponding patient clinical data of attributable in-hospital mortality (orange), total in-hospital mortality (gray), and septic shock (green).
(TIF)

**S1 Table. Multivariable logistic regression model of total in-hospital mortality in patients with *E. coli* bloodstream infections (BSI).** Covariates with p<0.15 in univariable logistic regression analyses were included in a multivariable logistic regression model. The final multivariable logistic regression model is shown here. P-values ≤0.05 are in bold.
(DOCX)

**S2 Table. Multivariable logistic regression model of attributable in-hospital mortality in patients with *E. coli* bloodstream infections (BSI).** Covariates with p<0.15 in univariable logistic regression analyses were included in a multivariable logistic regression model. The final multivariable logistic regression model is shown here. P-values ≤0.05 are in bold.
(DOCX)

## Author Contributions

**Conceptualization:** Orianna Poteete, Phillip Cox, Joshua T. Thaden.

**Data curation:** Orianna Poteete, Granger Sutton.

**Formal analysis:** Orianna Poteete, Granger Sutton, Lauren Brinkac, Thomas H. Clarke, Derrick E. Fouts, Joshua T. Thaden.

**Funding acquisition:** Granger Sutton, Derrick E. Fouts, Vance G. Fowler, Jr., Joshua T. Thaden.

**Investigation:** Orianna Poteete, Phillip Cox, Granger Sutton, Joshua T. Thaden.

**Methodology:** Orianna Poteete, Phillip Cox, Thomas H. Clarke, Derrick E. Fouts, Joshua T. Thaden.

**Project administration:** Felicia Ruffin, Joshua T. Thaden.

**Resources:** Orianna Poteete, Phillip Cox, Felicia Ruffin, Granger Sutton, Lauren Brinkac, Thomas H. Clarke, Derrick E. Fouts, Vance G. Fowler, Jr., Joshua T. Thaden.

**Software:** Orianna Poteete, Granger Sutton, Lauren Brinkac, Thomas H. Clarke, Derrick E. Fouts.

**Supervision:** Vance G. Fowler, Jr., Joshua T. Thaden.

**Validation:** Orianna Poteete, Phillip Cox.

**Visualization:** Orianna Poteete, Lauren Brinkac, Thomas H. Clarke, Derrick E. Fouts, Joshua T. Thaden.

**Writing – original draft:** Orianna Poteete.

**Writing – review & editing:** Orianna Poteete, Phillip Cox, Felicia Ruffin, Granger Sutton, Lauren Brinkac, Thomas H. Clarke, Derrick E. Fouts, Vance G. Fowler, Jr., Joshua T. Thaden.

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
