## [Decision Letter · Decision Letter 0]

3 Jun 2024

PONE-D-24-09625Serum Susceptibility of Escherichia coli and its Association with Patient Clinical OutcomesPLOS ONE

Dear Dr. Thaden,

Thank you for submitting your manuscript to PLOS ONE. After careful consideration, we feel that it has merit but does not fully meet PLOS ONE’s publication criteria as it currently stands. Therefore, we invite you to submit a revised version of the manuscript that addresses the points raised during the review process. =============================

The manuscript requires minor revision before it can be published - please respond to all the questions that have been raised by the two reviewers. 

We look forward to receiving your revised manuscript.

Kind regards,

Fiona J Radcliff

Academic Editor

PLOS ONE

Reviewers' comments:

Reviewer's Responses to Questions

**Comments to the Author**

1. Is the manuscript technically sound, and do the data support the conclusions?

Reviewer #1: Yes

Reviewer #2: Yes

2. Has the statistical analysis been performed appropriately and rigorously? 

Reviewer #1: Yes

Reviewer #2: No

3. Have the authors made all data underlying the findings in their manuscript fully available?

Reviewer #1: Yes

Reviewer #2: No

4. Is the manuscript presented in an intelligible fashion and written in standard English?

Reviewer #1: Yes

Reviewer #2: Yes

5. Review Comments to the Author

Reviewer #1: I believe that all experimental techniques have been well executed and statistical analyses applied appropriately. Although serum bacterial resistance and the clinical outcomes of mortality and septic shock appear to be unrelated, the authors have put this result in context by describing preceding literature studies and discussed the current study's limitations, while providing a high-throughput and efficient alternative approach for quantifying serum killing to the time-consuming classical method.

In terms of revisions, I suggest these minor points be addressed: (1) One thing that was not mentioned in the text was that the error margins for the flow method was much less than that for SBA's (see Fig 2), which in favour of this manuscript and could be suggested as an addition to the text. (2) Fig 1 labels "hours of work" which should really be described as "duration" since not every minute is being used as manual activity by the researcher. (3) Line 214: It was stated that the proportion of 50% serum-surviving bacteria was less than the 25% serum-surviving bacteria but is a mean proportion of 0.67 (50% serum) really significant compared to 0.68 (25% serum)?

Reviewer #2: Very nice paper. I answered no to question 2 due to lack of statistical comparison between methods used for determining survival in serum (Figure 2). I answered no to question 3 due to lack of information about how antibiotic resistance was determined and whether you refer to genotypic or phenotypic resistance. If these analyses were conducted as part of the previous study referenced, state that in methods and in-text. Once addressed, I am happy to check yes to those boxes.

My other recommendations as well as those made above are detailed below:

Recommendations:

Line 43/44 – ‘The complement system is a regulated network of proteins in the blood that is composed of three pathways’ to indicate that the three pathways are distinct, e.g. ‘The complement system is a regulated network of proteins in the blood with three potential pathways for activation’.

Line 48 – provide purpose of MAC complex to indicate that MAC insertion causes cell lysis

Line 103 – Remove line ‘This is the standard approach for performing an SBA’ and change to ‘As in the standard approach for performing an SBA, E. coli isolates were streaked onto … ‘ to continue into next sentence.

Line 180 – Consider performing bioinformatic analysis to determine bacterial phylogroup via Clermont typing or EzClermont (https://ezclermont.hutton.ac.uk/)- can run off PC or command line. The results would likely be interesting as you might find the isolates more resistant to serum belong to the B2/D phylogroups and those which are sensitive may belong to another phylogroup, indicating the infection was driven by immune status of the host, rather than virulence.

Line 210 – Perform statistical analysis of results from your SBA vs Flow methods to strengthen the argument that results are reproducible and comparable to the traditionally used method. ANOVA would be a good test. Likely, there would be no significance between the results for each isolate using the two different methods, which would reaffirm the efficacy and reliability of your Flow method.

Line 250 – you mention ‘bacterial characteristics’ in table, in reference to antibiotic resistance yet there is no indication in methods for how this was determined. Are you referring to phenotypic or genotypic resistance? If phenotypic, were AMR genes identified in the whole genome sequencing analysis? If the AMR experiments were performed on the isolates in the previous paper your group has published, ensure to reference that in the text as well as in methods e.g. 123 isolates were found to be MDR, as detailed in (ref) OR ‘as determined by the presence of genes or mutations associated with resistance’.

Line 269 – If possible, include sequence types on legend in supplementary figure regarding bacterial genotype. If phylogroup is determined as suggested above, include also.

Line 314 – Another study has explored the serum resistance and general characteristics of 20 E. coli bloodstream isolates associated with patient mortality in an Irish hospital - PMID: 26518234. The aforementioned study showed that in the E. coli isolates which displayed in vitro serum sensitivity, the patients had several co-morbidities often associated with poor outcomes. I would suggest looking for a similar trend among your isolates, to highlight that both bacterial (virulence) and host factors (health status) can play into morbidity and patient outcome. Additionally, I would include that study in your reference list.

Line 331 – I suggest correlating serum sensitive isolates to patient data as mentioned above. If not possible due to ethics, state that such a comparison would prove useful but was not possible due to ethics and consider referencing papers which did so.

6. PLOS authors have the option to publish the peer review history of their article (what does this mean?). If published, this will include your full peer review and any attached files.

Reviewer #1: No

Reviewer #2: **Yes: **Naoise McGarry

---

## [Author Response · Author response to Decision Letter 0]

25 Jun 2024

We have attached a separate letter detailing our response to Reviewers comments. This has also been pasted below.

Dear Editor and Reviewers:

Thank you for your positive responses to our manuscript submission. Enclosed is the revision of the manuscript submission PONE-D-24-09625 to be considered for publication in PLOS ONE. We thank you for your thoughtful comments on our manuscript. We have incorporated these changes into this revised version. We believe that this has resulted in an improved final version. The following summarizes how we responded to each provided comment.

Reviewer #1

I believe that all experimental techniques have been well executed and statistical analyses applied appropriately. Although serum bacterial resistance and the clinical outcomes of mortality and septic shock appear to be unrelated, the authors have put this result in context by describing preceding literature studies and discussed the current study's limitations, while providing a high-throughput and efficient alternative approach for quantifying serum killing to the time-consuming classical method. In terms of revisions, I suggest these minor points be addressed: 

1. One thing that was not mentioned in the text was that the error margins for the flow method was much less than that for SBA's (see Fig 2), which in favour of this manuscript and could be suggested as an addition to the text. 

Thank you for the suggestion. We have added this point to the Discussion. Lines 317-319 (in the “clean” version of the manuscript) now read:

“First, our novel flow cytometry-based approach for measuring E. coli serum susceptibility produced results similar to the gold standard study (traditional SBA) but with decreased overall time, hands-on bench work, and variation between biological replicates.”

2. Fig 1 labels "hours of work" which should really be described as "duration" since not every minute is being used as manual activity by the researcher. 

Agreed. Throughout the manuscript, “duration” has replaced the prior wording used to describe the time needed to complete the experiment. For example, the Figure 1 legend title now reads as follows at lines 136-137

“Figure 1. Workflow and duration differences between traditional serum bactericidal assay (SBA) and flow cytometry-based SBA methods.”

”

3. (3) Line 214: It was stated that the proportion of 50% serum-surviving bacteria was less than the 25% serum-surviving bacteria but is a mean proportion of 0.67 (50% serum) really significant compared to 0.68 (25% serum)?

Agreed. The difference between the two distributions is not significant. The following line was deleted:

“As expected, the 50% serum treatment distribution was shifted left relative to the 25% serum treatment curve as the higher serum percentage resulted in more bacterial death.” 

Reviewer #2

4. I answered no to question 2 due to lack of statistical comparison between methods used for determining survival in serum (Figure 2). 

As detailed in our response to comment 10 below, we have added statistical analyses (ANOVA tests) to verify that there were no statistically significant differences in measured serum susceptibility between the two methods.

5. I answered no to question 3 due to lack of information about how antibiotic resistance was determined and whether you refer to genotypic or phenotypic resistance. If these analyses were conducted as part of the previous study referenced, state that in methods and in-text. Once addressed, I am happy to check yes to those boxes.

Agreed. As in the response to comment 11 below, antibiotic susceptibility was determined by phenotypic testing by the Duke Clinical Microbiology Laboratory as part of routine clinical care. This point, as well as the definition of MDR, was added in lines 101-103:

“Antimicrobial susceptibility testing was performed by the Duke Clinical Microbiology Lab using standard techniques. The multidrug resistant (MDR) phenotype was defined as in Magiorakos et al. (24).”

6. Line 43/44 – ‘The complement system is a regulated network of proteins in the blood that is composed of three pathways’ to indicate that the three pathways are distinct, e.g. ‘The complement system is a regulated network of proteins in the blood with three potential pathways for activation’.

Agreed. Lines 43-44 now reads as follows:

“The complement system is a regulated network of proteins in the blood that is composed of three pathways: classical, lectin, and alternative.”

7. Line 48 – provide purpose of MAC complex to indicate that MAC insertion causes cell lysis

Agreed. Lines 46-48 now read:

“Activation of any pathway can result in formation of the membrane attack complex (MAC), a pore in the bacterial plasma membrane that triggers pathogen death through osmolysis.”

8. Line 103 – Remove line ‘This is the standard approach for performing an SBA’ and change to ‘As in the standard approach for performing an SBA, E. coli isolates were streaked onto … ‘ to continue into next sentence.

Agreed. This line has been removed and replaced with the suggested text. Lines 106-107 now read as follows:

“As in the standard approach for performing an SBA, E. coli isolates were streaked onto LB plates and incubated at 37˚C overnight (ON).”

9. Line 180 – Consider performing bioinformatic analysis to determine bacterial phylogroup via Clermont typing or EzClermont (https://ezclermont.hutton.ac.uk/)- can run off PC or command line. The results would likely be interesting as you might find the isolates more resistant to serum belong to the B2/D phylogroups and those which are sensitive may belong to another phylogroup, indicating the infection was driven by immune status of the host, rather than virulence.

Agreed. The phylogroup was determined for each E. coli isolate, and we performed statistical analyses to identify differences in serum susceptibility between the groups. This was added to the manuscript at lines 281-285:

“E. coli phylogroups included A (n=3 [2%]), B1 (n=7 [4%]), B2 (n=129 [70%]), C (n=1 [1%]), D (n=37 [20%]), and F (n=6 [3%]). Overall, there was no difference in serum resistance among phylogroups (25% serum: p=0.27; 50% serum: p=0.59). Similarly, pairwise comparisons (e.g., phylogroup B2 vs. all others) did not identify phylogroups associated with higher or lower serum resistance.”

10. Line 210 – Perform statistical analysis of results from your SBA vs Flow methods to strengthen the argument that results are reproducible and comparable to the traditionally used method. ANOVA would be a good test. Likely, there would be no significance between the results for each isolate using the two different methods, which would reaffirm the efficacy and reliability of your Flow method.

Agreed. We performed a ANOVA tests to verify that measured serum susceptibility results did not significantly differ after exposing bacterial isolates to either 25% serum or 50% serum. This data was added to the text (Lines 208-210), the Figure 2 legend, and to the Statistical Analysis section of the Methods at lines 148-152:

“Two-way analysis of variance (ANOVA) tests were used to validate our novel flow cytometry-based method relative to the traditional SBA. This test accounted for both the bacterial isolate (high, medium, and low serum susceptibility isolates were tested) and the method for determining serum susceptibility (flow cytometry-based versus traditional SBA).”

11. Line 250 – you mention ‘bacterial characteristics’ in table, in reference to antibiotic resistance yet there is no indication in methods for how this was determined. Are you referring to phenotypic or genotypic resistance? If phenotypic, were AMR genes identified in the whole genome sequencing analysis? If the AMR experiments were performed on the isolates in the previous paper your group has published, ensure to reference that in the text as well as in methods e.g. 123 isolates were found to be MDR, as detailed in (ref) OR ‘as determined by the presence of genes or mutations associated with resistance’.

Antibiotic susceptibility was determined by phenotypic testing by the Duke Clinical Microbiology Laboratory as part of routine clinical care. This point, as well as the definition of MDR, was added in lines 101-103:

“Antimicrobial susceptibility testing was performed by the Duke Clinical Microbiology Lab using standard techniques. The multidrug resistant (MDR) phenotype was defined as in Magiorakos et al. (24).”

12. Line 269 – If possible, include sequence types on legend in supplementary figure regarding bacterial genotype. If phylogroup is determined as suggested above, include also.

Agreed. Both the sequence types and phylogroups were added to Supplemental Figure 2.

13. Line 314 – Another study has explored the serum resistance and general characteristics of 20 E. coli bloodstream isolates associated with patient mortality in an Irish hospital - PMID: 26518234. The aforementioned study showed that in the E. coli isolates which displayed in vitro serum sensitivity, the patients had several co-morbidities often associated with poor outcomes. I would suggest looking for a similar trend among your isolates, to highlight that both bacterial (virulence) and host factors (health status) can play into morbidity and patient outcome. Additionally, I would include that study in your reference list.

Agreed. We examined the patient clinical factors associated with patient mortality. This data is shown in both Supplemental Tables 1 and 2, and has been added to the manuscript at lines 266-271:

“Clinical factors associated with increased total mortality in this cohort included increasing age (Odds ratio 1.04; 95% confidence interval 1.01-1.08; p=0.02), community-acquired/healthcare-associated infection (relative to hospital-acquired infection) (Odds ratio 0.36; 95% confidence interval 0.13-0.99; p=0.05), hemodialysis dependence (Odds ratio 5.43; 95% confidence interval 1.61-18.92; p<0.01), and an unknown source of BSI (Odds ratio 4.65;83 95% confidence interval 1.53-14.; p<0.01).”

In addition, the study (PMID 26518234) has been added to the Discussion (lines 336-337) and cited.

14. Line 331 – I suggest correlating serum sensitive isolates to patient data as mentioned above. If not possible due to ethics, state that such a comparison would prove useful but was not possible due to ethics and consider referencing papers which did so.

Agreed. We performed these analyses, though did not identify any patient or bacterial (i.e., antibiotic resistance) factors associated with E. coli serum susceptibility. This was added to the manuscript at lines 247-248:

“We did not identify any significant associations between serum susceptibility and the patient and bacterial variables in Table 1.”

Thank you for considering this manuscript for publication. All authors had full access to the data and take responsibility for the integrity and accuracy of the data. This manuscript has not been submitted or accepted elsewhere. All authors contributed to and approved the submitted version of the manuscript. The authors have declared that no competing interests exist.

---

## [Editor Report · Decision Letter 1]

16 Jul 2024

Serum Susceptibility of Escherichia coli and its Association with Patient Clinical Outcomes

PONE-D-24-09625R1

Dear Dr. Thaden,

We’re pleased to inform you that your manuscript has been judged scientifically suitable for publication and will be formally accepted for publication once it meets all outstanding technical requirements.

Kind regards,

Fiona J Radcliff

Academic Editor

PLOS ONE
---

## [Editor Report · Acceptance letter]

19 Jul 2024

PONE-D-24-09625R1 

PLOS ONE

Dear Dr. Thaden, 

I'm pleased to inform you that your manuscript has been deemed suitable for publication in PLOS ONE. Congratulations! Your manuscript is now being handed over to our production team.

Kind regards, 

on behalf of

Dr. Fiona J Radcliff 

Academic Editor

PLOS ONE